# Rationale and design of the THIRST Alert feasibility study: a pragmatic, single-centre, parallel-group randomised controlled trial of an interruptive alert for oral fluid restriction in patients treated with intravenous furosemide

Yang Chen [1,2] Anoop Shah,[1,2] Yogini Jani [3] Daniel Higgins,[2] Nausheen Saleem,[2] Kris Chafer,[2] Matthew Robert Sydes [4,5] Folkert W Asselbergs,[1,6] R Thomas Lumbers[1]

For numbered affiliations see end of article.

**Correspondence to**
Dr R Thomas Lumbers;
t.lumbers@ucl.ac.uk

## ABSTRACT

**Introduction** Acute heart failure (HF) is a major cause of unplanned hospitalisation characterised by excess body water. A restriction in oral fluid intake is commonly imposed on patients as an adjunct to pharmacological therapy with loop diuretics, but there is a lack of evidence from traditional randomised controlled trials (RCTs) to support the safety and effectiveness of this intervention in the acute setting.

This study aims to explore the feasibility of using computer alerts within the electronic health record (EHR) system to invite clinical care teams to enrol patients into a pragmatic RCT at the time of clinical decision-making. It will additionally assess the effectiveness of using an alert to help address the clinical research question of whether oral fluid restriction is a safe and effective adjunct to pharmacological therapy for patients admitted with fluid overload.

**Methods and analysis** THIRST (Randomised Controlled **T**rial within the electronic **H**ealth record of an **I**nterruptive alert displaying a fluid **R**estriction **S**uggestion in patients with the treatable **T**rait of congestion) Alert is a single-centre, parallel-group, open-label pragmatic RCT embedded in the EHR system that will be conducted as a feasibility study at an National Health Service (NHS) hospital in London. The clinical care team will be invited to enrol suitable patients in the study using a point-of-care alert with a target sample size of 50 patients. Enrolled patients will then be randomised to either restricted or unrestricted oral fluid intake. Two primary outcomes will be explored (1) the proportion of eligible patients enrolled in the study and (2) the mean difference in oral fluid intake between randomised groups. A series of secondary outcomes are specified to evaluate the effectiveness of the alert, adherence to the randomised treatment allocation and the quality of data generated from routine care, relevant to the outcomes of interest.

## STRENGTHS AND LIMITATIONS OF THIS STUDY

⇒ THIRST Alert is a pragmatic randomised controlled trial where all elements of trial conduct are embedded within the routine care clinical pathway.
⇒ The intervention and programme theory has been codesigned with a multidisciplinary team including physicians, nurses, clinical informatics officers and patient representatives.
⇒ A proportionate consent model has been approved for use to help with the practicality of trial conduct.
⇒ Uncertainty remains regarding data quality in relation to the ascertainment of study outcomes from routine hospital care records.

**Ethics and dissemination** This study was approved by Riverside Research Ethics Committee (Ref: 22/LO/0889) and will be published on completion.
**Trial registration number** NCT05869656.

## INTRODUCTION

Acute heart failure (HF) is an important cause of unplanned hospitalisation and is characterised by excess body water, also known as congestion.[1 2] Despite being a common clinical problem, there is a lack of high-quality evidence to guide drug treatment and non-pharmacological measures.[3] Restriction of oral fluid intake is commonly used as an adjunct to pharmacological therapy but its effectiveness is uncertain and may exacerbate symptoms of thirst, as suggested by the limited randomised controlled trials (RCTs), which have been conducted to date.[4–6] These previous trials achieved only small differences

in oral fluid intake, which may reflect the Hawthorne effect whereby participation in a clinical trial influences fluid intake in both groups.[7]

Patients and clinicians identify the need for better treatment of fluid overload as a key priority, and the use of oral fluid restriction also highlighted a gap in evidence in clinical practice guidelines.[2 8 9] To address this evidence gap and some of the challenges encountered in previous studies, we designed a pragmatic RCT embedded in the electronic health record (EHR) system to evaluate the safety and effectiveness of oral fluid restriction in acute HF.

## Patient population

Fluid overload may be caused by liver or kidney dysfunction,[10 11] as well as by HF, with similar evidence gaps for congestion management in such circumstances.[12–14] However, a reliance on evaluating interventions for the immediate management of fluid overload only where the underlying cause is established may impact the generalisability of any findings. Given that standardised clinical scores for congestion are not well validated or adopted in practice,[15 16] we used the prescription of two consecutive doses of intravenous furosemide within the first 48 hours of admission as evidence of a physician-assessed 'treatable trait' of fluid congestion, to define the study population.[17]

At our proposed study site, more than one consecutive dose of intravenous furosemide was administered during the first 48 hours of admission in 1537 unplanned admissions between April 2019 and October 2022. In 56% of these admissions with available supporting data, oral fluid restriction was recommended (online supplemental appendix 1). The use of prescribing intention to identify the study population enabled digital enrolment into the study based on a discrete triggering event. To our knowledge, there has been no previous RCT conducted in all patients treated for congestion with intravenous furosemide in the acute care setting.

## Pragmatic research

There are many evidence gaps in routine clinical practice that have not been addressed by conventional RCTs, which may be prohibitively expensive. Pragmatic clinical trials (PCTs) offer a potential solution, particularly relevant to the evaluation of low-risk interventions where clinical equipoise may exist. By integrating a randomisation procedure into routine care pathways, automating case identification through the EHR, engaging the clinical care team for patient recruitment and using routinely collected data for trial outcomes, there is the potential to perform large-scale trials at low-cost and low burden to patients and their caregivers. Many questions remain, however, about how best to deliver PCTs embedded in EHR systems, including the design of physician-facing alerts and different models of consent. For our study, we employ an interruptive alert to provide the routine care team with an invitation to enrol eligible patients, and use

a proportionate process for consent that can be delivered by the same team.

PCTs have the potential to enable evidence generation during routine care, helping to conduct more efficient and representative trials.[18] Through examining the comparative effectiveness of existing interventions that have demonstrable variation in practice, many evidence gaps may be addressed. The use of oral fluid restriction in patients who are treated pharmacologically for fluid overload in the setting of acute unplanned care represents one such gap.

The THIRST (Randomised Controlled **T**rial within the electronic **H**ealth record of an **I**nterruptive alert displaying a fluid **R**estriction **S**uggestion in patients with the treatable **T**rait of congestion) Alert trial, therefore aims to examine (1) whether interruptive alerts are an effective means to engage the usual care team to enrol patients into a pragmatic RCT and (2) for enrolled patients, whether the randomised assignment to either oral fluid restriction to 1 L per day or no oral fluid restriction leads to a difference in documented oral fluid intake. The evidence generated will be used to inform the design of subsequent multicentre outcomes-driven PCTs embedded in EHR systems.

## METHODS AND ANALYSIS

### Study design

THIRST Alert is a single-centre, parallel-group, open-label pragmatic RCT embedded in the EHR system that will be conducted as a feasibility study.[19] The study will be delivered entirely at University College Hospital, UCLH, a digitally mature National Health Service (NHS) hospital.[20] Enrolment started in May 2023 and the estimated trial completion date is December 2023. Given the low-risk nature of the intervention, a verbal, opt-out consent model is used. The study protocol has been prepared with reference to the Standard Protocol Items: Recommendations for Interventional Trials (SPIRIT) statement.[21] The trial is registered at ClinicalTrials.gov (NCT05869656).

### Study procedures

An overview of the study procedures is given in figure 1. The trial was designed to be integrated into the EHR system EPIC (Epic, Epic Systems, Verona, Wisconsin, USA). The screening and identification of eligible patients, display of interruptive alerts to physicians, randomisation and outcome ascertainment are all provisioned within EPIC.[22] Education events, internal communication and email correspondence were delivered to members of the routine care team to raise awareness of the study but no formal cointerventions were undertaken. The conduct and analysis of the trial are overseen by a trial management group that conceived a programme theory for the alert with relevant multidisciplinary input (online supplemental appendix 2,3). The enrolment invitation and randomised treatment

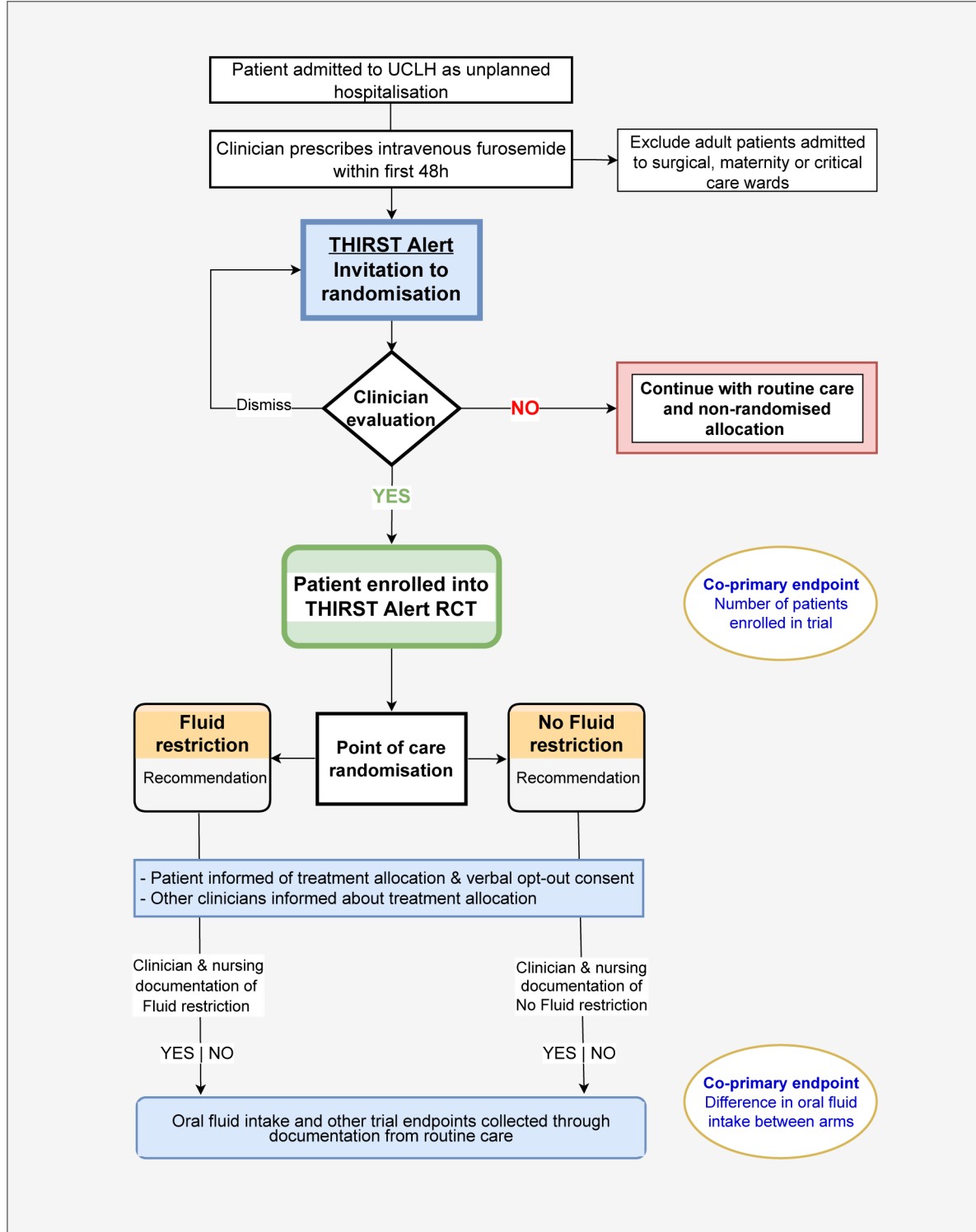

**Figure 1** THIRST Alert trial patient flow. RCT, randomised controlled trial; THIRST, Randomised Controlled **T**rial within the electronic **H**ealth record of an **I**nterruptive alert displaying a fluid **R**estriction **S**uggestion in patients with the treatable **T**rait of congestion; UCLH, University College London Hospital.

recommendations are classified as complex interventions according to the Medical Research Council framework.[23]

The trial is designed to maximise the generalisability and applicability of the evidence generated and is highly pragmatic according to the PRagmatic Explanatory Continuum Indicator Summary-2 (PRECIS-2) framework (figure 2).[24]

### Study participants

Adults of 18 years or older who were administered more than one dose of intravenous furosemide within the first

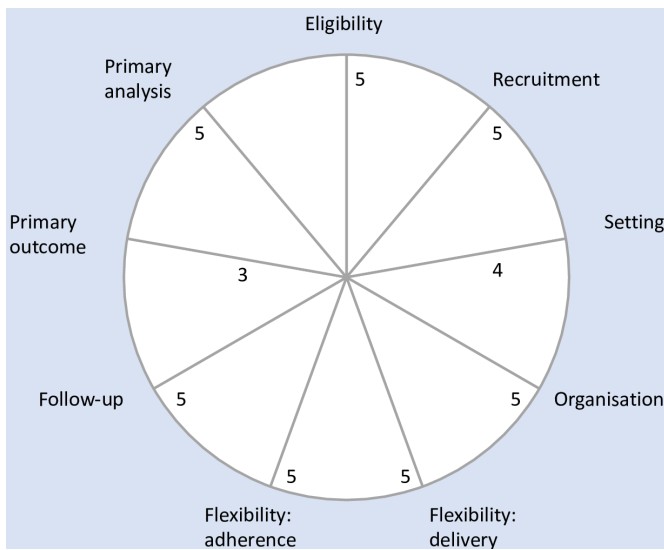

**Figure 2** PRECIS-2 score for the THIRST Alert trial. Each of the nine domains are scored on a 5-point Likert continuum (from 1=very explanatory 'ideal conditions' to 5=very pragmatic 'usual care conditions'). PRECIS-2 score, The PRagmatic Explanatory Continuum Indicator Summary-2 score; THIRST, Randomised Controlled **T**rial within the electronic **H**ealth record of an **I**nterruptive alert displaying a fluid **R**estriction **S**uggestion in patients with the treatable **T**rait of congestion.

48 hours of an unplanned admission were eligible for the study. Patients admitted to a surgical, maternity or critical care ward were excluded (table 1).

### The interruptive alert inviting usual care team to enrol eligible patients

Any member of the clinical team may receive an interruptive alert (online supplemental appendix 4 figure 1A) during their interaction with EPIC for an eligible patient, informing them of the study and inviting them to enrol the patient. The clinician can click the appropriate button on the alert to enrol, decline to enrol or defer the decision regarding enrolment. The alert is non-coercive, and the decision about patient enrolment is at the discretion of the treating clinician. We anticipate that physicians will enrol patients in cases where they judge that there is clinical equipoise as to whether oral fluid restriction is likely to benefit the patient. The alert was designed and tested with EPIC application specialists in an iterative manner, integrating feedback from end-users on the wording of the design.

### The randomised treatment intervention

After enrolment of patients in the study, the routine care team receives a recommendation regarding oral fluid intake and assignment is determined by point of care randomisation. The randomised treatment allocation is only presented to the clinical team after they click 'yes' to the THIRST alert, rather than displayed immediately, to minimise the possibility of differential recruitment to the treatment arms of the trial. For patients allocated to the intervention group, the routine care team received a recommendation for a target oral fluid restriction of 1 L per day; for patients allocated to the control group, the care team received a recommendation for unrestricted oral fluid intake (online supplemental appendix 4 figure 1B,C). The oral fluid restriction target was chosen based on previous RCTs and acceptability in clinical practice.[4–6] The randomised treatment recommendation for both groups additionally advises the clinician to: (1) complete an EPIC order for fluid balance monitoring; (2) document the treatment allocation in the clinical notes; (3) inform the patient and the nursing team and (4) provide the patient with a participant information sheet (PIS).

### Randomisation and blinding

Subjects will be randomised 1:1 using an internal random number rule implemented using a standard EPIC randomisation tool used in other PCTs.[25] No block randomisation or additional covariate balancing will be undertaken. The study is open label with patients and their care teams being made aware of the treatment allocation.

| **Table 1** Eligibility criteria for thirst alert study participants | |
|---|---|
| **Study participants** | **Eligibility criteria** |
| Patients | **Inclusion**<br>▶ Aged 18 years and over<br>▶ Prescribed intravenous furosemide during the first 48 hours of their admission as a regular prescription rather than a one-off dose.<br>▶ Assessed as being suitable for inclusion in the trial by the responsible clinical team, that is, fluid restriction deemed to be in equipoise.<br>**Exclusion**<br>▶ Surgical or obstetric wards<br>▶ Critical care wards |
| Prescribing clinicians: physicians (consultants and junior doctors) | Any clinician with prescribing rights and who prescribes more than one dose of intravenous furosemide within 48 hours of admission, during the trial recruitment period. |
| Nursing staff | A separate order is triggered for enrolled patients and only presented to nursing staff who access their patient record to help facilitate fluid balance documentation. |

## Study outcomes

### Primary outcomes

1. Number of eligible patients randomised.
2. Difference in oral fluid intake between intervention and control arms.

### Secondary outcomes

1. Adherence to randomised treatment recommendation.
2. Proportion of alerts resulting in clinical orders for nursing care.
3. Oral fluid intake.
4. Net fluid balance.
5. Weight change after randomisation.
6. Length of stay.
7. Frequency of blood test measurements of renal function.
8. Prescription of diuretic medications.
9. Daily change in creatinine.
10. Patient-reported outcome measures.

These outcomes were chosen based on clinical relevance and the ability to detect the possibility of attributable harm during an inpatient admission.

### Sample size and power calculation

No formal sample size calculation was performed given the primary outcomes pertain to the feasibility of using an EHR alert aimed at the routine care team to (1) enrol eligible patients into an RCT of oral fluid restriction and (2) cause and sustain the allocated clinical treatment effect, as measured by documented oral fluid intake. This is in keeping with the role of feasibility studies used by other groups.[26] Our target sample size was 50 patients, based on the predicted number of eligible patients at the study site during the recruitment period.

### Data collection, curation and storage

Baseline characteristics and study outcomes will be extracted from the routine care record. Information on the frequency of the interruptive alert and staff responses to alerts will be recorded in EPIC. We will also extract primary and secondary International Classification of Diseases (ICD)-10 diagnosis codes assigned to the clinical episode by the clinical coding team. Study data will be transferred to a secure research environment within UCLH NHS Foundation Trust as pseudoanonymised electronic case report forms (eCRFs). No data will leave the UCLH NHS system. Patients who opt out or withdraw from the trial will have their decision documented on the eCRF and their data will not be included in the main analysis.

### Statistical analysis plan

Baseline characteristics of study participants will be summarised using percentages or means and SD as appropriate and compared using t tests for continuous variables (expressed as mean±SD) and the $\chi^2$ tests or the Fisher's exact tests for categorical variables (expressed as count and percentage). Secondary endpoints for intervention and control groups will be compared using $\chi^2$ test for binary outcomes, and t-test for continuous outcomes. For subgroup analyses, linear or logistic regression will be used to determine the effects of treatment allocation according to HF status. No adjustment or imputation will be used for missing data. For this feasibility study, we considered a sample of 20 participants enrolled as a minimum number acceptable for feasibility and a daily mean difference of 250 mL of fluid intake between treatment arms as clinically relevant. Progression to a multi-centre clinical outcomes trial will be contingent on a series of stop/go criteria based on achieving the primary outcome targets and on secondary outcomes relating to data quality.

## MONITORING AND EVALUATION

The trial will report serious adverse events (SAEs) that are attributable to the study through the EHR, in accordance with the safety reporting processes and sponsor policy.[27] The Trial Management Group will review recruitment rates, SAEs and any substantial amendments to the protocol.

### Patient and public involvement

Patients were integrally involved in the design of the study as previously reported.[28] In brief this involved a patient participation event where options for consent and outcome measures were explored which subsequently informed the study design. In accordance with the National Institute for Health Research guidance on coproduction,[29] one participant (KC) served as a patient advisor to the trial and reviewed all trial materials.

### Ethics and dissemination

This study protocol was approved by the London Riverside Research Ethics Committee (Ref: 22/LO/0889) and sponsored by University College London (Ref: 151938). A verbal opt-out model of consent was adopted based on the negligible risk associated with the intervention. Patients are informed of their study participation and of their treatment allocation and can opt-out of the study or decline to follow the recommendation for oral fluid restriction at any time. A copy of the sample PIS is provided to patients (online supplemental appendix 5). The data management for the trial conforms to NHS Information standards DCB0129 and DCB0160 and was approved by University College London Hospitals NHS Foundation Trust (UCLH) Digital Services Safety board.[30] The alert is not classified as a separate piece of software but is incorporated within the Epic EHR, which is registered as a class I medical device by the UK Medicines and Healthcare products Regulatory Agency.[31 32] The study findings will be disseminated through publication in open-access peer-reviewed journals and through the provision of reports for patients and clinicians involved in the study. As a feasibility study, the findings shared will include trial evaluation, and whether alerts which 'nudge' clinicians to recruit patients into pragmatic, low-risk comparative effectiveness trials,[33] are a scalable way to conduct efficient RCTs.[18] The THIRST Alert feasibility

study will share insights that inform the design and execution of future integrated approaches to evidence generation from routine care.

## Author affiliations
[1]Institute of Health Informatics, University College London, London, UK
[2]Clinical and Research Informatics Unit, NIHR UCLH Biomedical Research Centre, University College London Hospitals NHS Foundation Trust, London, UK
[3]Centre for Medicines Optimisation Research & Education - CMORE, University College London Hospitals NHS Foundation Trust, London, UK
[4]Institute of Clinical Trials and Methodology, Medical Research Council Clinical Trials Unit at University College London, London, UK
[5]Health Data Research UK, London, UK
[6]Department of Cardiology, Amsterdam Cardiovascular Sciences, Amsterdam University Medical Centre, University of Amsterdam, Amsterdam, Netherlands

**Acknowledgements** We are grateful to clinical colleagues, Inaki Elizondo, Pamela Stephenson, Sian Roberts-Walsh, Alexander Tranter and Ali Hosin for assistance with the study design. We thank Steve Harris and Pier Lambiase for comments on the study protocol and to members of the digital safety committee with the safe implementation of this study.

**Contributors** YC, AS, MRS, FWA and RTL conceived the study idea. YC and RTL wrote the first draft of the protocol. YJ, DH and NS helped to design, test and review the study alert. KC reviewed and edited the patient-facing study materials. All authors reviewed the manuscript and contributed to revisions.

**Funding** This work was supported by the National Institute for Health and Care Research (NIHR) UCLH Biomedical Research Centre (BRC), award number NIHR-INF-1873.

**Disclaimer** The study funder did not play any role in the design of the study.

**Competing interests** RTL has received funding from Pfizer for unrelated research, under the Innovative Target Exploration Network programme.

**Patient and public involvement** Patients and/or the public were involved in the design, or conduct, or reporting, or dissemination plans of this research. Refer to the Methods section for further details.

**Patient consent for publication** Not applicable.

**Provenance and peer review** Not commissioned; externally peer reviewed.

**ORCID iDs**
Yang Chen http://orcid.org/0000-0001-6032-3387
Yogini Jani http://orcid.org/0000-0001-5927-5429
Matthew Robert Sydes http://orcid.org/0000-0002-9323-1371

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
