## [Reviewer comments · BMJ Open]

ARTICLE DETAILS

TITLE (PROVISIONAL)	Rationale and design of the THIRST Alert feasibility study: a pragmatic, single centre, parallel group randomised controlled trial of an interruptive alert for oral fluid restriction in patients treated with intravenous furosemide
AUTHORS	Chen, Yang; Shah, Anoop; Jani, Yogini; Higgins, Daniel; Saleem, Nausheen; Chafer, Kris; Sydes, Matthew; Asselbergs, Folkert; Lumbers, R Thomas

VERSION 1 – REVIEW

REVIEWER	Seki, Tomotsugu Kyoto Prefectural University of Medicine, Department of Cardiovascular Medicine
REVIEW RETURNED	21-Oct-2023

GENERAL COMMENTS	The study protocol by Chen et al. presents a novel approach that is both relevant and well-articulated in the manuscript. While the content is compelling, there are several areas that require clarification and modification to enhance the clarity, coherence, and validity of the study. 1. Title: The current study title is not entirely reflective of its content. It would be more accurate if the title incorporated terms like "feasibility" or "pilot" to convey the true nature of the trial.2. Abbreviation: Abbreviations "UCLH" and "CTIMP" appear without prior introduction. Please spell out upon its first mention.3. Sample Size Justification: The manuscript lacks a formal sample size calculation. It's crucial to understand the basis for the chosen sample size. Could the authors elucidate how they determined the number of patients required for this study?4. Consistency in Patient Recruitment Details: There is an inconsistency in the reported recruitment duration and numbers across various sections. The study design indicates an eight-month recruitment period (May to December 2023), while the statistical analysis plan cites a three-month period for 20 participants. Additionally, ClinicalTrials.gov lists 50 patients for recruitment. The authors should reconcile these discrepancies and state the correct recruitment details.5. Rationale for Randomization: The justification for randomization in this feasibility study remains unclear. Could the authors provide insights into why randomization is vital, especially regarding the interpretation of the second co-primary outcome (Mean difference in oral fluid intake for 48 hours post-randomization)?
--

	6. Randomization Method: Given the small sample size, imbalance between two groups is possible. Why did the authors opt against using techniques like block or covariate balancing randomization which could mitigate this risk? 7. Statistical Analysis Clarification: The statistical analysis plan mentions a "minimum daily mean difference of 250ml of fluid intake" as being clinically relevant. What underpins this determination? I hope these suggestions serve to enhance the presentation and rigor of the study. Looking forward to the authors' revisions. Reference Thabane L, Lancaster G. A guide to the reporting of protocols of pilot and feasibility trials. Pilot and Feasibility Studies. 2019 Feb 28;5(1):37.
--	--

REVIEWER	Philip, Keir National Heart and Lung Institute
REVIEW RETURNED	18-Nov-2023

GENERAL COMMENTS	Many thanks for inviting me to review this study protocol. It is clearly an important study both in relation to the larger THIRST RCT and more broadly in building an evidence base for components of pragmatic clinical trials. I was impressed by the importance of the research question and the trial's design. Studies such as this have real potential to improve patient care. The points I raise for clarification are minor: The title should state this is a feasibility study. Please add absolute numbers of patients to the statement 'the first 48 hours of admission in 1 in 25 of unplanned admissions between April 2019 to April 2022.' As this would be a useful guide for likely recruitment. Do we know if anyone is being put on oral fluid restriction while on oral furosemide? Most probably not, or very few, but worth mentioning if data available. As fluid intake going to be measured in those who are not on oral fluid restriction the Hawthorn effect is likely to still be relevant which will need to be discussed in the subsequent paper reporting the findings. Are you confident in the accuracy of documented fluid intake recordings in this setting? Has this been assessed? Is there some way of checking its accuracy? Please provide a reference/justification for the statement 'We consider a minimum daily mean difference of 250ml of fluid intake to be clinically relevant.' Secondary outcome 9 should include route of diuretics also – i.e. when the patients are switched to oral if this is possible. Secondary outcome 11 VAS scales should be provided. Are they validated, what is the scale, do they have any prompts etc.
--

VERSION 1 – AUTHOR RESPONSE

Reviewer: 1

Dr. Tomotsugu Seki, Kyoto Prefectural University of Medicine

Comments to the Author:

The study protocol by Chen et al. presents a novel approach that is both relevant and well-articulated in the manuscript. While the content is compelling, there are several areas that require clarification and modification to enhance the clarity, coherence, and validity of the study.

Many thanks for reviewing our paper your comments. We address your individual points below.

1. Title: The current study title is not entirely reflective of its content. It would be more accurate if the title incorporated terms like "feasibility" or "pilot" to convey the true nature of the trial.

We have updated the title to include the word 'feasibility':

"Rationale and design of the THIRST Alert feasibility study: a pragmatic, parallel group randomised controlled trial of an interruptive alert for oral fluid restriction in patients admitted to hospital and treated with intravenous furosemide"

2. Abbreviation: Abbreviations "UCLH" and "CTIMP" appear without prior introduction. Please spell out upon its first mention.

We have updated this in the manuscript. We have chosen to remove 'non-CTIMP' entirely and simplified the sentence to

'The trial will report serious adverse events (SAEs) that are attributable to the study through the EHR, in accordance with the safety reporting processes and sponsor policy.'

3. Sample Size Justification: The manuscript lacks a formal sample size calculation. It's crucial to understand the basis for the chosen sample size. Could the authors elucidate how they determined the number of patients required for this study?

As a feasibility study, there is no target sample size required to 'power the primary outcome' given that the primary outcomes chosen were the number of patients enrolled and the difference in oral fluid intake. We have followed advice from NIHR:

<https://www.nihr.ac.uk/documents/nihr-research-for-patient-benefit-rfpb-programme-guidance-on-applying-for-feasibility-studies/20474>

The previous reference supplied:

'Eldridge SM, Lancaster GA, Campbell MJ, Thabane L, Hopewell S, Coleman CL, et al. (2016) Defining Feasibility and Pilot Studies in Preparation for Randomised Controlled Trials: Development of a Conceptual Framework. PLoS ONE 11(3): e0150205. doi:10.1371/journal.pone.0150205'

specifies that there is overlap between feasibility and pilot studies and highlighted this in a systematic review of the literature. We elected to use the term feasibility given our more novel method of patient recruitment in routine care.

We have expanded the 'Sample size and power calculation' section as follows:

"No formal sample size calculation was performed given the primary outcomes pertain to the feasibility of using an EHR alert aimed at the routine care team to (i) enrol eligible patients into an RCT of oral fluid restriction and (ii) cause and sustain the allocated clinical treatment effect, as measured by documented oral fluid intake. This is in keeping with the role of feasibility studies used by other groups. [additional new reference inserted –

Blatch-Jones AJ, Pek W, Kirkpatrick E, et al. Role of feasibility and pilot studies in randomised controlled trials: a cross-sectional study. *BMJ Open* 2018;8:e022233. doi:10.1136/bmjopen-2018-022233]

Our target sample size was 50 patients, based on the predicted number of eligible patients at the study site during the recruitment period."

4. Consistency in Patient Recruitment Details: There is an inconsistency in the reported recruitment duration and numbers across various sections. The study design indicates an eight-month recruitment period (May to December 2023), while the statistical analysis plan cites a three-month period for 20 participants. Additionally, ClinicalTrials.gov lists 50 patients for recruitment. The authors should reconcile these discrepancies and state the correct recruitment details.

Many thanks for highlighting this inconsistency. In reply:

1. Our target sample size during the recruitment window of 6 months is 50 patients and we have added text to specify this.

2. In the statistical analysis plan, we have updated the text to state that 20 is the minimum number of enrolled patients which defines feasibility of the recruitment method used in the study. This was pre-specified by the trial management group and agreed by our study sponsor. This was the lower bound that we felt defined feasibility in terms of representing a proportion of patients being looked after by routine care clinicians who felt that there was justification for enrolment into an RCT of fluid restriction (ie there was clinical equipoise present).

3. Regarding the total recruitment period, the clinical trials.gov entry is correct. The original protocol stated 3 months but during the initial recruitment period, we made an adjustment to the triggering rules which display the alert, due to concerns about over-sensitivity. This resulted in a 1 month period where the alert was switched off and no longer visible to the clinicians.

We subsequently sought an amendment from our study sponsor and the Health Research Authority to extend the total period active of recruitment to 6 months as a result of fixing the alert and external factors in the UK (NHS doctor strikes). The amendment was approved and this therefore explains the dates on the clinicaltrials.gov record. Please note, the study completion date includes the possibility of the last patient being followed up within hospital for up to 30 days – therefore explaining the total 8 month period.

5. Rationale for Randomization: The justification for randomization in this feasibility study remains unclear. Could the authors provide insights into why randomization is vital, especially regarding the interpretation of the second co-primary outcome (Mean difference in oral fluid intake for 48 hours post-randomization)?

The use of randomisation is important to robustly determine the second co-primary outcome of whether patients enrolled by the first alert are then being managed in their intended treatment allocation (supplemental figure 1). The randomisation occurs at the point of care, after the clinician clicks 'enrol'. If the first alert already displayed a treatment allocation and the patient was 'pre-randomised' there could have been differential recruitment as clinicians may have been biased by the pre-made allocation and failed to enrol patients in a selected manner. We have added the following sentence in quotation marks.

The randomised treatment intervention

After enrolment of patients in the study, the routine care team receives a recommendation regarding oral fluid intake and assignment is determined by point of care randomisation.

" The randomised treatment allocation is only presented to the clinical team after they click 'yes' to the THIRST alert, rather than displayed immediately, to minimise the possibility of differential recruitment to the treatment arms of the trial."

As part of the trial evaluation, we will also be verifying whether randomisation was possible and worked in the manner we intended.

6. Randomization Method: Given the small sample size, imbalance between two groups is possible. Why did the authors opt against using techniques like block or covariate balancing randomization which could mitigate this risk?

1:1 randomisation using a simple internal number rule was the choice of point of care randomisation made available to the study team by the EHR vendor. While for larger alert-based studies, this simple randomisation did not cause imbalance, we accept there is a greater risk for this given our smaller sample. We will discuss this as a limitation in our results paper.

7. Statistical Analysis Clarification: The statistical analysis plan mentions a "minimum daily mean difference of 250ml of fluid intake" as being clinically relevant. What underpins this determination?

There is no robust evidence to determine what is clinically relevant. Of the 3 traditional RCTs conducted of fluid restriction (References 4-6), only 2 (Machado d'Almeida et al and Travers et al) reported the difference in oral intake between treatment. These were approximately 300ml and 400ml respectively. This is the authorship groups opinion is that 250ml is relevant and we set a lower bound based on our small sample size and reliance of data documentation from routine care.

I hope these suggestions serve to enhance the presentation and rigor of the study. Looking forward to the authors' revisions.

Reference

Thabane L, Lancaster G. A guide to the reporting of protocols of pilot and feasibility trials. *Pilot and Feasibility Studies*. 2019 Feb 28;5(1):37.

Reviewer: 2

Dr. Keir Philip, National Heart and Lung Institute

Comments to the Author:

Many thanks for inviting me to review this study protocol. It is clearly an important study both in relation to the larger THIRST RCT and more broadly in building an evidence base for components of pragmatic clinical trials. I was impressed by the importance of the research question and the trial's design. Studies such as this have real potential to improve patient care.

Many thanks for reviewing our paper your comments. We address your individual points below.

The points I raise for clarification are minor:
The title should state this is a feasibility study.

We have updated the title accordingly

Please add absolute numbers of patients to the statement 'the first 48 hours of admission in 1 in 25 of unplanned admissions between April 2019 to April 2022.' As this would be a useful guide for likely recruitment.

We have used raw numbers and adjusted the end date to October 2022 for consistency with the supplement. The 1 in 25 figure has been removed as the denominator used to calculate the fraction relied upon data from NHS Digital which we will need to re-check.

Do we know if anyone is being put on oral fluid restriction while on oral furosemide? Most probably not, or very few, but worth mentioning if data available.

We currently do not have robust data for this, we will know on what day patients were switched to oral and will report this in the main trial finding.

As fluid intake going to be measured in those who are not on oral fluid restriction the Hawthorn effect is likely to still be relevant which will need to be discussed in the subsequent paper reporting the findings.

We plan to discuss the Hawthorn effect in detail for the main trial report and evaluation

Are you confident in the accuracy of documented fluid intake recordings in this setting? Has this been assessed? Is there some way of checking its accuracy?

We are preparing a parallel paper on data documentation quality of fluid balance in routine care during the pre-trial observation period and will also report this in the main trial. This will include frequency of intake measurements, data quality and how they vary according to location and provider.

Please provide a reference/justification for the statement 'We consider a minimum daily mean difference of 250ml of fluid intake to be clinically relevant.'

There is no robust evidence to determine what is clinically relevant. Of the 3 traditional RCTs conducted of fluid restriction (References 4-6), only 2 (Machado d’Almeida et al and Travers et al) reported the difference in oral intake between treatment. These were approximately 300ml and 400ml respectively. This is the authorship groups opinion is that 250ml is relevant and we set a lower bound based on our small sample size and reliance of data documentation from routine care.

Secondary outcome 9 should include route of diuretics also – i.e. when the patients are switched to oral if this is possible.

Secondary outcome 11 VAS scales should be provided. Are they validated, what is the scale, do they have any prompts etc.

As specified by the editor, we have updated the wording of our outcome measures to ensure the same wording as the ‘outcome measures’ data field on the clinicaltrials.gov entry. This has resulted in some of the outcomes losing granular detail e.g., ‘Prescription of diuretic medications’ instead of ‘Duration and Dose of diuretic medications’. The detail is available in the “measure description” data field on the clinicaltrials.gov entry and the formatting we used was based on requests by their administrative team who manage that database.

Reviewer: 1

Competing interests of Reviewer: N/A

Reviewer: 2

Competing interests of Reviewer: None.

VERSION 2 – REVIEW

REVIEWER	Seki, Tomotsugu Kyoto Prefectural University of Medicine, Department of Cardiovascular Medicine
REVIEW RETURNED	06-Dec-2023

GENERAL COMMENTS	The authors have made appropriate revisions to the manuscript in accordance with the reviewer's suggestions. I believe that the current version of the manuscript is now suitable for publication.
--

REVIEWER	Philip, Keir
-----------------	--------------

	National Heart and Lung Institute
REVIEW RETURNED	06-Dec-2023
GENERAL COMMENTS	All my comments have been addressed.

VERSION 2 – AUTHOR RESPONSE